# INDUCTIVE LINK PREDICTION IN KNOWLEDGE GRAPHS USING PATH-BASED NEURAL NETWORKS

## ABSTRACT

Link prediction is a crucial research area in knowledge graphs, with many downstream applications. In many real-world scenarios, inductive link prediction is required, where predictions have to be made among unseen entities. Embedding-based models usually need fine-tuning on new entity embeddings, and hence are difficult to be directly applied to inductive link prediction tasks. Logical rules captured by rule-based models can be directly applied to new entities with the same graph typologies, but the captured rules are discrete and usually lack generosity. Graph neural networks (GNNs) can generalize topological information to new graphs taking advantage of deep neural networks, which however may still need fine-tuning on new entity embeddings. In this paper, we propose SiaILP, a path-based model for inductive link prediction using light-weight siamese neural networks. Our model only depends on relation and path embeddings, which can be generalized to new entities without fine-tuning. Experiments show that our model achieves several new state-of-the-art performances in link prediction tasks using inductive versions of WN18RR, FB15k-237, and Nell995.

## 1 INTRODUCTION

Link prediction is a crucial task in network analysis that involves predicting the existence or likelihood of a connection between two nodes in a network (Bordes et al., 2013; Sun et al., 2019; Yang et al., 2014). It has various applications, including social network analysis (Valente et al., 2015), recommendation systems (Zhang et al., 2019), and biological network analysis (Pavlopoulos et al., 2011). In recent years, there has been a growing interest in the inductive approach to link prediction, which involves predicting links between nodes that are not present in the training knowledge graph (Teru et al., 2019; Chen et al., 2021; Lin et al., 2022; Mai et al., 2021).

Compared to the transductive link prediction scenario where models are trained and evaluated on a fixed set of entities, the inductive link prediction scenario is more challenging. This is because inductive link prediction models need to generalize to unseen entities for evaluation (Teru et al., 2019). Due to this reason, traditional embedding-based link prediction models (Bordes et al., 2013; Sun et al., 2019; Yang et al., 2014; Trouillon et al., 2016; Dettmers et al., 2018) cannot be directly applied to the inductive scenario, where new entities do not possess trained embeddings. In contrast, many rule-based link prediction models (Meilicke et al., 2018; Yang et al., 2017a; Sadeghian et al., 2019) explicitly capture entity-invariant topological structures from the training knowledge graph. The learned rules can then be applied to unseen entities with the same topological structures. However, the learned rules are discrete and usually suffer from sparsity, making rule-based models lack of generosity (Teru et al., 2019).

Graph neural networks (GNNs) (Veličković et al., 2018; Kipf & Welling, 2017) can implicitly capture the topological structures of a graph into network weights, and hence can be generalized to larger and more complicated graphs containing unseen entities. A series of GNNs-based inductive link prediction models have been developed in recent years, achieving promising performances (Vashishth et al., 2020; Teru et al., 2019; Mai et al., 2021; Lin et al., 2022; Pan et al., 2022). However, most GNNs-based models still rely on entity embeddings (Vashishth et al., 2020; Schlichtkrull et al., 2018), which can be problematic when fine-tuning on new entity embeddings is forbidden.

In this paper, we present novel inductive link prediction models based on light-weight siamese neural networks. Our models are path-based in order to capture entity-invariant topological structures from

a knowledge graph. To be specific, our connection-based model predicts the target relation using the connecting paths between two entities, while our subgraph-based model predicts the target relation using the out-reaching paths from two entities. Both of our models exclude entity embeddings, which therefore can be directly applied to new knowledge graphs of the same topological structures without any fine-tuning. Experiments show that our models achieve several new state-of-the-art performances on the inductive versions (Teru et al., 2019) of the benchmark link prediction datasets WN18RR (Toutanova & Chen, 2015), FB15K-237 (Toutanova et al., 2015) and Nell-995 (Xiong et al., 2017).

Our models apply **sia**mese neural network for **i**nductive **l**ink **p**rediction, therefore are named as SiaILP. Briefly speaking, we recognize two advantages from our models:

1. **Strictly inductive**: Our models are path-based excluding entity embeddings, which can be directly applied to new entities for link prediction without fine-tuning.

2. **New state-of-the-art**: We apply our models to the inductive versions (Teru et al., 2019) of link prediction datasets WN18RR (Toutanova & Chen, 2015), FB15K-237 (Toutanova et al., 2015) and Nell-995 (Xiong et al., 2017). Experiments show that our models achieve several new state-of-the-art performances compared to other benchmark models.

In the following section, we will briefly introduce related work for link prediction. Then, we will present our models in Section 3, after describing some basic concepts of link prediction. After that, experimental results will be provided in Section 4. Finally, we conclude this paper in Section 5.

## 2 RELATED WORK

In this section, we would introduce benchmark models published in the recent years for both transductive link prediction and inductive link prediction.

**Transductive Link Prediction:** The field of knowledge graph representation learning has gained significant attention in the last decade. Inspired by the success of word embeddings in language modeling (Mikolov et al., 2013; Pennington et al., 2014), various link prediction models have been created based on entity and relation embeddings, including TransE (Bordes et al., 2013), Distmult (Yang et al., 2014), ComplEx (Trouillon et al., 2016), RotatE (Sun et al., 2019), and ConvE (Dettmers et al., 2018). However, these models often treat each triplet independently and do not consider the topological structure of the knowledge graph. Recently, graph neural networks (GNNs), such as graph convolutional networks (GCNs) (Kipf & Welling, 2017) and graph attention networks (GATs) (Veličković et al., 2018), have been designed to capture global topological and structural information inherent in knowledge graphs. Models like CompGCN (Vashishth et al., 2020), RGCN (Schlichtkrull et al., 2018), WGCN (Shang et al., 2019), and VR-GCN (Ye et al., 2019) apply GCNs to the link prediction problem using the topological structure from a knowledge graph, achieving new state-of-the-art results on benchmark datasets such as WN18RR (Toutanova & Chen, 2015), FB15K-237 (Toutanova et al., 2015), and Nell-995 (Xiong et al., 2017).

**Inductive Link Prediction:** Rule-based models like RuleN (Meilicke et al., 2018), Neural-LP (Yang et al., 2017b), and DRUM (Sadeghian et al., 2019) use logical and statistical approaches to capture knowledge graph structures and topology as explicit rules for inductive link prediction, but these rules often lack generality. In contrast, graph neural networks (GNNs) implicitly capture knowledge graph structures into network parameters, offering greater adaptability. GraIL (Teru et al., 2019) is a typical application of GNNs on inductive link prediction, along which the inductive versions of WN18RR (Toutanova & Chen, 2015), FB15K-237 (Toutanova et al., 2015) and Nell-995 (Xiong et al., 2017) datasets are presented as benchmark evaluation on inductive link prediction models.

Following GraIL, CoMPILE (Mai et al., 2021) uses communicative message-passing GNNs to extract directed enclosing subgraphs for each triplet in inductive link prediction tasks. Graph convolutional network (GCN)-based models like ConGLR (Lin et al., 2022), INDIGO (Liu et al., 2021), and LogCo (Pan et al., 2022) are also applied to inductive link prediction. ConGLR combines context graphs with logical reasoning, LogCo integrates logical reasoning and contrastive representations into GCNs, and INDIGO transparently encodes the input graph into a GCN for inductive link prediction. Lastly, TACT focuses on relation-corrupted inductive link prediction using a relational correlation graph (RCG) (Chen et al., 2021).

Beyond these models, NBFNet (Zhu et al., 2021) takes advantages of both traditional path-based model and graph neural networks for inductive link prediction, which is very similar to our model: Instead, we combine traditional path-based models with light-weight siamese neural networks other than GNNs for inductive link prediction, which is described in the next section. All the inductive link prediction models mentioned in this section will serve as our baselines.

## 3 METHODOLOGY

In this section, we will first describe the problem of link prediction and its related concepts. Then, we will introduce the structures of our connection-based model as well as our subgraph-based model for inductive link prediction. After that, we will provide details on our recursive path finding algorithm.

### 3.1 DESCRIPTION ON THE PROBLEM AND CONCEPTS

A knowledge graph can be denoted as $\mathcal{G} = (\mathcal{E}, \mathcal{R}, \mathcal{T})$, where $\mathcal{E}$ and $\mathcal{R}$ represent the set of *entities* and *relations*, respectively. A *triple* $(s, r, t)$ in the triple set $\mathcal{T}$ indicates that there is a relation $r$ from the *source entity* $s$ to the *target entity* $t$. We say that there is a *path* $r_1 \wedge r_2 \wedge \cdots \wedge r_k$, from entity $s$ to $t$, if there are entities $e_1, \cdots e_{k-1}$ in $\mathcal{E}$ such that $(s, r_1, e_1), (e_1, r_2, e_2), \cdots, (e_{k-1}, r_k, t)$ are known triples in $\mathcal{T}$. We use the letter $p$ to denote a path. We use $|\mathcal{E}|$, $|\mathcal{R}|$ and $|\mathcal{T}|$ to denote the number of entities, number of relations, and number of triples in $\mathcal{G}$, respectively. In this paper, we recognize a path $p$ only by its relation sequence $r_1 \wedge r_2 \wedge \cdots \wedge r_k$. The on-path entities $e_1, \cdots e_{k-1}$ are ignored.

Then, *knowledge graph completion*, or *link prediction*, means that given the known graph $\mathcal{G}$, we need to predict (in probability) whether an unknown triple $(s, r, t)$ is correct. To be specific, *transductive link prediction* guarantees that both $s$ and $t$ exist in $\mathcal{E}$, while *inductive link prediction* assumes either $s$ or $t$ to be unknown. For inductive link prediction, an *inference knowledge graph* $\mathcal{G}_{inf} = (\mathcal{E}_{inf}, \mathcal{R}, \mathcal{T}_{inf})$ is given with new entity set $\mathcal{E}_{inf}$ and new triples $\mathcal{T}_{inf}$, but the relation set $\mathcal{R}$ and graph topology are invariant. Then, the model will predict the correctness of $(s, r, t)$ based on $\mathcal{G}_{inf}$.

Besides, in most knowledge graphs, the relation $r$ is directed. For instance, *(monkey, has_part, tail)* is a known triple in WordNet while *(tail, has_part, monkey)* is not (Socher et al., 2013). Then, the inverse relation $r^{-1}$ can be defined for each relation $r \in \mathcal{R}$, so that $(t, r^{-1}, s)$ is a known triple whenever $(s, r, t)$ is. Accordingly, we can expand the relation set to be $\mathcal{R} \cup \mathcal{R}^{-1}$ with $\mathcal{R}^{-1} = \{r^{-1}\}_{r \in \mathcal{R}}$; and the triple set becomes $\mathcal{T} \cup \mathcal{T}^{-1}$ with $\mathcal{T}^{-1} = \{(t, r^{-1}, s)\}_{(s,r,t) \in \mathcal{T}}$. Hence, the *inverse-added knowledge graph* will be $\tilde{\mathcal{G}} = (\mathcal{E}, \mathcal{R} \cup \mathcal{R}^{-1}, \mathcal{T} \cup \mathcal{T}^{-1})$. To minimize confusion, we still use the symbol $\mathcal{G}$.

In this paper, we always work with inverse-added knowledge graphs. We always expand the relation set to $\mathcal{R} \cup \mathcal{R}^{-1}$ and triple set to $\mathcal{T} \cup \mathcal{T}^{-1}$. In this way, a path can be formed by connecting both initial and inverse relations. For example, we can have a path $p = r_1 \wedge r_2^{-1} \wedge r_3$ from $s$ to $t$, where $r_1, r_3 \in \mathcal{R}$ and $r_2^{-1} \in \mathcal{R}^{-1}$ (or in other words $r_2 \in \mathcal{R}$). Moreover, given a path $p = r_1 \wedge \cdots \wedge r_k$ from $s$ to $t$, we can always obtain the *inverse path* $p^{-1} = r_k^{-1} \wedge \cdots \wedge r_1^{-1}$, which is from $t$ to $s$.

### 3.2 STRUCTURE OF THE CONNECTION-BASED MODEL

Suppose we have an inverse-added knowledge graph $\mathcal{G} = (\mathcal{E}, \mathcal{R} \cup \mathcal{R}^{-1}, \mathcal{T} \cup \mathcal{T}^{-1})$. Inspired by the Word2Vec model (Mikolov et al., 2013), we apply the input and output embeddings to the relations in $\mathcal{G}$. That is, we will generate both the input embedding $\mathbf{v}_r$ and the output embedding $\tilde{\mathbf{v}}_r$ for each relation $r \in \mathcal{R} \cup \mathcal{R}^{-1}$. This gives us the input embedding matrix $(\mathbf{v}_1, \cdots, \mathbf{v}_{2|\mathcal{R}|}) \in \mathbb{R}^{D \times 2|\mathcal{R}|}$ and the output embedding matrix $(\tilde{\mathbf{v}}_1, \cdots, \tilde{\mathbf{v}}_{2|\mathcal{R}|}) \in \mathbb{R}^{D \times 2|\mathcal{R}|}$, where $D$ is the dimension size of the embedding and $|\mathcal{R}|$ is the number of initial relations.

Given two entities $s$ and $t$ in the graph $\mathcal{G}$, suppose there are three different paths, $p_1 = r_1 \wedge \cdots \wedge r_k$, $p_2 = r_1' \wedge \cdots \wedge r_{k'}'$ and $p_3 = r_1'' \wedge \cdots \wedge r_{k''}''$, from $s$ to $t$. Here, the path length and the composing relations of each path can be different. Then, the framework of our connection-based model is very simple: Given $p_1$, $p_2$ and $p_3$ from $s$ to $t$, the model will predict (in probability) whether the triple $(s, r, t)$ is correct with respect to a target relation $r \in \mathcal{R}$. Figure 1 (a) briefly indicates this framework.

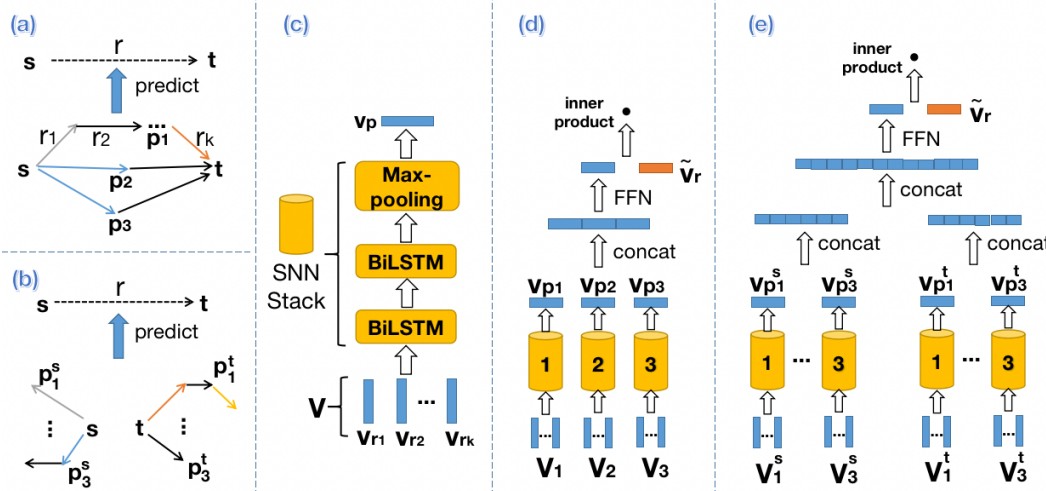

Figure 1: The architectures of our proposed models. (a): Given the connection paths from $s$ to $t$, our connection-based model will predict the target relation $r$. (b): Given out-reaching paths from $s$ and $t$ respectively, our subgraph-based model will predict the target relation $r$. (c): The architecture of one stack in our siamese neural network. (d): The architecture of our connection-based model. (e): The architecture of our subgraph-based model.

Before continuing, note that we only need to consider $r \in \mathcal{R}$, i.e. an initial relation, as our target relation. This is because predicting the correctness of $(s, r^{-1}, t)$ given paths $p_1$, $p_2$ and $p_3$ from $s$ to $t$ is equivalent to predicting that of $(t, r, s)$ given the inverse paths $p_1^{-1}$, $p_2^{-1}$ and $p_3^{-1}$ from $t$ to $s$.

We obtain embedding sequences $\mathbf{V}_1 = (\mathbf{v}_1, \cdots, \mathbf{v}_k)$, $\mathbf{V}_2 = (\mathbf{v}'_1, \cdots, \mathbf{v}'_{k'})$ and $\mathbf{V}_3 = (\mathbf{v}''_1, \cdots, \mathbf{v}''_{k''})$ for paths $p_1$, $p_2$, and $p_3$, respectively. Each $\mathbf{v}_i$ represents the input embedding of the relation at the corresponding path position. We construct a connection-based model using a Siamese Neural Network (SNN) framework. This involves three stacks of deep neural networks, sharing the same parameters and weights during training. Stack one takes $\mathbf{V}_1$ as input, stack two takes $\mathbf{V}_2$, and stack three takes $\mathbf{V}_3$.

Each stack, as shown in Figure 1 1 (c), consists of two layers of bi-directional LSTM (bi-LSTM) layers (Hochreiter & Schmidhuber, 1997) and one max-pooling layer, denoted as $f_{LSTM_1}$, $f_{LSTM_2}$ and $f_{max}$, respectively. Within a bi-LSTM layer, the hidden vectors from forward and backward LSTM networks are concatenated at each time step. Suppose $(\mathbf{h}_1, \cdots, \mathbf{h}_k) = f_{LSTM_2}(f_{LSTM_1}((\mathbf{v}_1, \cdots, \mathbf{v}_k)))$ are the output hidden vectors from the second bi-LSTM layer when $\mathbf{V}_1$ is the input. We will make sure $\mathbf{h}_i \in \mathbb{R}^D$ for $i = 1, \cdots, k$. Then, we will do dimension-wise max-pooling: $\mathbf{h}_{max} = f_{max}(\mathbf{h}_1, \cdots, \mathbf{h}_k)$ to obtain $\mathbf{h}_{max} \in \mathbb{R}^D$, where each dimension $h_{max}^d = \max(h_1^d, \cdots, h_k^d)$ for $d = 1, \cdots, D$.

We use $\mathbf{h}_{max}^{\mathbf{V}_1}$ to denote the output vector from the stack implementing on $\mathbf{V}_1$. Similarly, we will obtain $\mathbf{h}_{max}^{\mathbf{V}_2}$ and $\mathbf{h}_{max}^{\mathbf{V}_3}$ from the stacks implementing on $\mathbf{V}_2$ and $\mathbf{V}_3$, respectively. After that, we concatenate these three output vectors into one vector $\mathbf{h}_{concat} \in \mathbb{R}^{3D}$. Then, a feed forward network $f_{FFN}$ is applied: $\mathbf{h}_{rep} = f_{FFN}(\mathbf{h}_{concat})$, where $\mathbf{h}_{rep} \in \mathbb{R}^D$ is the vector representation of the three paths $p_1$, $p_2$ and $p_3$. One can regard $\mathbf{h}_{rep}$ as the 'connection embedding' for $p_1, p_2, p_3$.

Finally, given a target relation $r \in \mathcal{R}$, we will obtain its output embedding $\tilde{\mathbf{v}}_r$. We will further normalize the magnitude of $\mathbf{h}_{rep}$ and $\tilde{\mathbf{v}}_r$ to unit length: $||\mathbf{h}_{rep}||_2 = ||\tilde{\mathbf{v}}_r||_2 = 1$, where $||\cdot||_2$ is the L-2 norm. Then, we calculate $\mathbf{h}_{rep} \cdot \tilde{\mathbf{v}}_r$, the inner product between $\mathbf{h}_{rep}$ and $\tilde{\mathbf{v}}_r$, which is the final output value of the connection-based model. We denoted this output value as $P((s, r, t)|p_1, p_2, p_3)$ or $P_{(s,r,t)}^{connection}$, which is the evaluated probability for $(s, r, t)$ to be correct, given paths $p_1$, $p_2$ and $p_3$ between $s$ and $t$. Figure 1 (d) describes the architecture of the connection-based model.

However, there could be less than three different paths between two entities $s$ and $t$. Then, we will need the subgraph-based model for link prediction.

### 3.3 STRUCTURE OF THE SUBGRAPH-BASED MODEL

Given two entities $s$ and $t$ in the graph $\mathcal{G}$, suppose there are three different paths $p_1^s$, $p_2^s$ and $p_3^s$ from $s$ to some other entities; also, suppose there are three different paths $p_1^t$, $p_2^t$ and $p_3^t$ from $t$ to some other entities. Then, we build another model to predict (in probability) whether the triple $(s, r, t)$ is correct with respect to a target relation $r \in \mathcal{R}$, given $p_1^s$, $p_2^s$, $p_3^s$, $p_1^t$, $p_2^t$ and $p_3^t$. Figure 1 (b) briefly indicates this framework. Since the out-reaching paths essentially represents the subgraph around an entity, we refer to this model as the subgraph-based model.

Again, we only need to consider $r \in \mathcal{R}$, an initial relation, as our target relation in the triple $(s, r, t)$. This is because predicting the correctness of $(s, r^{-1}, t)$ given $p_1^s, p_2^s, p_3^s$ from $s$ and $p_1^t, p_2^t, p_3^t$ from $t$ is equivalent to predicting that of $(t, r, s)$ given $p_1^t, p_2^t, p_3^t$ from $t$ and $p_1^s, p_2^s, p_3^s$ from $s$.

Similarly, we obtain embedding sequences $\mathbf{V}_1^s$, $\mathbf{V}_2^s$, $\mathbf{V}_3^s$, $\mathbf{V}_1^t$, $\mathbf{V}_2^t$ and $\mathbf{V}_3^t$ for paths $p_1^s, p_2^s, p_3^s, p_1^t, p_2^t$ and $p_3^t$, respectively as in the connection-based model. The source and target paths share the same input embedding matrix, which is independent from the embedding matrix in the connection-based model. Again, we build SNN stacks to perform on the embedding sequences: Given $\mathbf{V}_1^s$, $\mathbf{V}_2^s$ and $\mathbf{V}_3^s$ as input, each stack in the SNN will path them through two bi-LSTM layers and one max-pooling layer to obtain $\mathbf{h}_{max}^{\mathbf{V}_1^s} \in \mathbb{R}^D$, $\mathbf{h}_{max}^{\mathbf{V}_2^s} \in \mathbb{R}^D$ and $\mathbf{h}_{max}^{\mathbf{V}_3^s} \in \mathbb{R}^D$, respectively. Then, $\mathbf{h}_{concat}^s \in \mathbb{R}^{3D}$ is obtained by concatenating these three vectors, which is the output of the SNN. Note that there is no topological difference between source entity subgraph and target entity subgraph. Hence, the same SNN is performed on $\mathbf{V}_1^t$, $\mathbf{V}_2^t$ and $\mathbf{V}_3^t$ to obtain $\mathbf{h}_{concat}^t \in \mathbb{R}^{3D}$. Then, we further concatenate $\mathbf{h}_{concat}^s$ and $\mathbf{h}_{concat}^t$ into vector $\mathbf{h}_{concat}^{(s,t)} \in \mathbb{R}^{6D}$. After that, a feed forward network $f_{FFN}$ will be applied: $\mathbf{h}_{rep}^{(s,t)} = f_{FFN}(\mathbf{h}_{concat}^{(s,t)})$, where $\mathbf{h}_{rep}^{(s,t)} \in \mathbb{R}^D$ can be viewed as the 'subgraph embedding' of $p_1^s, p_2^s, p_3^s, p_1^t, p_2^t$ and $p_3^t$.

Finally, given $r \in \mathcal{R}$, we normalize the magnitude of $\mathbf{h}_{rep}^{(s,t)}$ and $\tilde{\mathbf{v}}_r$ to unit length. Here, $\tilde{\mathbf{v}}_r$ is the output embedding of $r$, independent from the connection-based model. Then, we calculate the inner product between $\mathbf{h}_{rep}^{(s,t)}$ and $\tilde{\mathbf{v}}_r$, which is the final output of the subgraph-based model as described in Figure 1 (e). We denote this value as $P((s, r, t)|p_1^s, p_2^s, p_3^s, p_1^t, p_2^t, p_3^t)$ or $P_{(s,r,t)}^{subgraph}$, which is the evaluated probability for $(s, r, t)$ to be correct, given paths $p_1^s, p_2^s, p_3^s$ from $s$ and $p_1^t, p_2^t, p_3^t$ from $t$.

We can see that both our models are path-based, depending only on the topological structure in a knowledge graph. No entity embeddings are involved in our models.

### 3.4 RECURSIVE PATH-FINDING ALGORITHM

Given an entity $s$ in the graph $\mathcal{G}$, we use a recursive algorithm to find out-reaching paths from $s$. Intuitively, if $s$ reaches $t$ via path $p$, we will further extend $p$ by reaching out to each direct neighbor of $t$ recursively. Here is a more detailed description:

Suppose at the current step, we possess a path $p = r_1 \wedge \cdots \wedge r_k$ starting from s and ending at another entity $t$. Also, suppose the on-path entities are $\{s, e_1, \cdots, e_{k-1}, t\}$, which are unique so that the path is acyclic. Also, we set the upper bound on the length of a path to be $\mathbf{L}$. Also, we denote $C_l^s$ to be the number of recursions already performed from $s$ on paths of length $l$. Then, we set the upper bound on $C_l^s$ to be $\mathbf{C}$, for $l = 1, 2, \cdots, \mathbf{L} - 1$.

We define $Q_s$ to be the *qualified entity set* given the initial entity $s$. If entity $t$ at current recursion step does not belong to $Q_s$, we will not record the discovered path $p$. In this paper, $Q_s$ can be the entire entity set $\mathcal{E}$, or the set containing all the direct neighbors of $s$, or the set containing only one given entity $t_0$.

Define $N_t^s$ to be the number of discovered paths from $s$ to $t$. Again, in this paper, paths are only differed from each other by their relation sequences, rather than their on-path entities. We stop recording discovered paths from $s$ to $t$ when $N_s^t$ reaches $\mathbf{N}$, our pre-defined upper bound. Also, suppose the direct neighbors ("one-hop" connections) of $t$ are $\{t_1', \cdots, t_n'\}$, where each $t_i'$ is connected with $t$ via a triple $(t, r_i', t_i')$. Again, $r_i'$ may be either an initial or an inverse relation.

Finally, we will decide whether to continue the recursion for each direct neighbor $t_i'$ of the current entity $t$. That is, if the path length $|p| < \mathbf{L}$, the current number of recursions $C_{|p|}^s < \mathbf{C}$, and the direct

neighbor $t'_i$ does not belong to the current on-path entities $\{s, e_1, \cdots, e_{k-1}, t\}$, we will proceed to the next recursion step with respect to source entity $s$, path $p' = r_1 \wedge \cdots \wedge r_k \wedge r'_i$, target entity $t'_i$ and on-path entities $\{s, e_1, \cdots, e_{k-1}, t, t'_i\}$. In the meanwhile, $C^s_{|p|}$ will increase by 1. Here, on-path entities are considered to guarantee an acyclic path for next recursion.

Again, this path-finding algorithm depends on the starting entity $s$, which is summarized below.

---

**Algorithm 1:** Recursive Path-finding from entity $s$

---

**Initialize L, C, N.**
**Func** $(p = r_1 \wedge \cdots \wedge r_k, t, \{s, e_1, \cdots, e_{k-1}, t\}, Q_s)$:
**if** $t \in Q_s$ **and** $N^s_t < $ **N then**
   Record the path $p$ from $s$ to $t$;
   $N^s_t \longleftarrow N^s_t + 1$.
**end**
**for** $r'_i$, $t'_i$ in the direct neighbor of $t$ **do**
   **if** $|p| < $ **L and** $C^s_{|p|} < $ **C and** $t'_i \notin \{s, e_1, \cdots, e_{k-1}, t\}$ **then**
      **Func** $(p = r_1 \wedge \cdots \wedge r_k \wedge r'_i, t'_i, \{s, e_1, \cdots, e_{k-1}, t, t'_i\}, Q_s)$;
      $C^s_{|p|} \longleftarrow C^s_{|p|} + 1$.
   **end**
**end**
**Run Func** $(p = \emptyset, s, \{s\}, Q_s)$.

---

Here, we use **Func**$(x, y, \cdots)$ to represent a function with input $x$, $y$, etc. Also, $p = \emptyset$ means that the path $p$ starts from empty, or none, or length-zero. If $s$ reaches out to a direct neighbor $t$ by $r$, then $p$ recursively becomes $p = r$.

In the next section, we will introduce how to implement our models and path-finding algorithms on inductive link prediction datasets, as well as the performances of our model on these datasets.

## 4 EXPERIMENTAL RESULTS

In this section, we will first introduce the commonly used datasets for inductive link prediction model evaluation, as well as other published models providing baselines for comparison. Then, we will introduce the training methods and evaluation metrics we applied. The performances of our models are provided right after. Finally, an ablation study will be conducted to see if the existing structure reaches an optimism.

### 4.1 DATASETS AND BASELINE MODELS

We work on the benchmark datasets for inductive link prediction proposed with the GraIL model (Teru et al., 2019), which are derived from WN18RR (Toutanova & Chen, 2015), FB15k-237 (Toutanova et al., 2015), and NELL-995 (Xiong et al., 2017). Each of the WN18RR, FB15k-237 and NELL-995 datasets are further developed into four different versions for inductive link prediction. Each version of each dataset contains a training graph and an inference graph, whereas the entity set of the two graphs are disjoint. Detailed statistics on the number of entities, triples and relation types of the datasets are summarized in many papers, such as (Teru et al., 2019; Chen et al., 2021; Lin et al., 2022). For simplicity, we do not repeat the statistics in this paper again.

We adapt benchmark inductive link prediction models published in recent years as baselines in this paper. They are Neural-LP from (Yang et al., 2017a), RuleN from (Meilicke et al., 2018), DRUM from (Sadeghian et al., 2019), GraIL from (Teru et al., 2019), R-GCN from (Schlichtkrull et al., 2018), CoMPILE from (Mai et al., 2021), ConGLR from (Lin et al., 2022), INDIGO from (Liu et al., 2021), NBFNet from (Zhu et al., 2021), LogCo from (Pan et al., 2022) and TACT from (Chen et al., 2021). Performances of these models will be given in subsection 4.4.

## 4.2 TRAINING PROTOCOLS

Suppose we obtain the inverse-added knowledge graph $\mathcal{G} = (\mathcal{E}, \mathcal{R} \cup \mathcal{R}^{-1}, \mathcal{T} \cup \mathcal{T}^{-1})$ for training. For the connection-based SiaILP model, we implement the recursive path-finding algorithm on each entity $s \in \mathcal{E}$ to discover paths from $s$ to its direct neighbors. Specifically, we use $\mathbf{L} = 10$, $\mathbf{C} = 20000$, $\mathbf{N} = 50$, and set $Q_s$ to be the direct neighbors of $s$. For more details on this algorithm, please refer to subsection 3.4. We repeat this algorithm ten times for each $s \in \mathcal{E}$ to enrich the discovered paths.

Afterwards, we train the connection-based model using contrastive learning (negative sampling) with the discovered paths. We randomly select three paths $p_1$, $p_2$, and $p_3$ from $s$ to $t$ and provide their corresponding input embedding sequences $\mathbf{V}_1$, $\mathbf{V}_2$, and $\mathbf{V}_3$ to the connection-based model. We also provide the model with the output embedding of the target relation, which is either the ground-truth relation $r$ in a training triple $(s, r, t)$ (where $Q_s$ being the direct neighbor of $s$ ensures the existence of such a triple), or a randomly selected relation $r' \in \mathcal{R}$. The connection-based model calculates the inner product as described in subsection 3.2, with a label of 1 for the true relation $r$ and 0 for the random relation $r'$.

Here is the training strategy for the subgraph-based model: For each triple $(s, r, t) \in \mathcal{T}$, we randomly select two additional entities $s'$ and $t'$ from $\mathcal{E}$ and a random relation $r' \in \mathcal{R}$. We apply the recursive path-finding algorithm separately from $s$, $t$, $s'$, and $t'$, using $\mathbf{L} = 3$, $\mathbf{C} = 20000$, and $\mathbf{N} = 50$. However, for comprehensive subgraph representation, we set $Q_s$, $Q_t$, $Q_{s'}$, and $Q_{t'}$ to be the entire set $\mathcal{E}$. Similarly, the training is based on contrastive learning, which creates four triples: $(s, r, t)$, $(s, r', t)$, $(s, r, t')$, and $(s', r, t)$. For each triple, we randomly select three out-reaching paths from each entity as input paths. The model calculates the inner product as described in section 3.3, with a label of 1 for a true triple and 0 for a corrupted triple.

We set the dimension of each relation embedding to be $D = 300$ in both models. We set the number of hidden units to be $H = 150$ in each forward and backward LSTM layers in all siamese neural network stacks. The learning rate is $10^{-5}$, the batch size is 32 and the training epoch is 10 across all models on all datasets. The models are trained on an Apple M1 Max CPU.

## 4.3 EVALUATION METRICS

We apply both classification metric and ranking metric to evaluate the performance of our model. For classification metric, we use the area under the precision-recall curve (AUC-PR) following GraIL (Teru et al., 2019). That is, we replace the source or target entity of each test triple with a random entity to form a negative triple. Then, we score the positive test triples with an equal number of negative triples to calculate AUC-PR.

For the ranking metric, however, there seems to be two different settings. The first setting is purposed in GraIL (Teru et al., 2019) and followed by CoMPILE (Mai et al., 2021), ConGLR (Lin et al., 2022), INDIGO (Liu et al., 2021), NBFNet (Zhu et al., 2021) and LogCo (Pan et al., 2022), where each test triple is ranked among other 50 negative triples whose source or target entities are replaced by random entities. Accordingly, Hits@10 (the rate of true test triples ranked top-10 in all performed rankings) is calculated with respect to all test triples. We refer to this setting as **entity-corrupted ranking**. This setting coincides with that in the AUC-PR metric.

The second setting is proposed both in TACT (Chen et al., 2021) and INDIGO (Liu et al., 2021), where each test triple is ranked among other negative triples whose relation is replaced by other relations in the graph. Accordingly, Hits@1 is calculated with respect to all test triples in the paper presenting TACT (Chen et al., 2021), while Hit@3 is calculated in the paper presenting INDIGO (Liu et al., 2021). Here, the number of candidate triples depends on the number of relations in the graph. We refer to this setting as **relation-corrupted ranking**. To comprehensively evaluate our models, we apply both settings as our ranking metrics.

## 4.4 PERFORMANCES

The AUC-PR and entity-corrupted Hits@10 scores of Neural-LP, RuleN and DRUM are obtained from (Teru et al., 2019), while the relation-corrupted Hits@1 score of these three models are obtained from (Chen et al., 2021). The AUC-PR and entity-corrupted Hits@10 scores of TACT are

| Model | WN18RR | | | | FB15K-237 | | | | Nell-995 | | | |
|---|---|---|---|---|---|---|---|---|---|---|---|---|
| | v1 | v2 | v3 | v4 | v1 | v2 | v3 | v4 | v1 | v2 | v3 | v4 |
| Neural-LP | 86.02 | 83.78 | 62.90 | 82.06 | 69.64 | 76.55 | 73.95 | 75.74 | 64.66 | 83.61 | 87.58 | 85.69 |
| DRUM | 86.02 | 84.05 | 63.20 | 82.06 | 69.71 | 76.44 | 74.03 | 76.20 | 59.86 | 83.99 | 87.71 | 85.94 |
| RuleN | 90.26 | 89.01 | 76.46 | 85.75 | 75.24 | 88.70 | 91.24 | 91.79 | 84.99 | 88.40 | 87.20 | 80.52 |
| GraIL | 94.32 | 94.18 | 85.80 | 92.72 | 84.69 | 90.57 | 91.68 | 94.46 | 86.05 | 92.62 | 93.34 | 87.50 |
| TACT | 95.43 | 97.54 | 87.65 | 96.04 | 83.15 | 93.01 | 92.10 | 94.25 | 81.06 | 93.12 | 96.07 | 85.75 |
| CoMPILE | 98.23 | 99.56 | 93.60 | 99.80 | 85.50 | 91.68 | 93.12 | 94.90 | 80.16 | 95.88 | 96.08 | 85.48 |
| ConGLR | **99.58** | **99.67** | 93.78 | **99.88** | 85.68 | 92.32 | 93.91 | 95.05 | 86.48 | 95.22 | 96.16 | **88.46** |
| LogCo | 99.43 | 99.45 | **93.99** | 98.75 | **89.74** | 93.65 | **94.91** | 95.26 | **91.24** | **95.96** | **96.28** | 87.81 |
| SiaILP solo | 79.04 | 78.17 | 76.39 | 70.96 | 88.03 | **94.95** | 92.75 | **95.42** | 83.58 | 87.65 | 91.22 | 81.98 |
| SiaILP hybrid | 84.23 | 88.54 | 83.94 | 84.50 | 88.64 | 93.39 | 92.81 | 93.20 | 76.35 | 88.20 | 89.88 | 81.03 |

Table 1: The AUC-PR metric values (in %) of inductive link prediction on twelve dataset versions. The best score is in **bold** and the second best one is underlined.

| Model | WN18RR | | | | FB15K-237 | | | | Nell-995 | | | |
|---|---|---|---|---|---|---|---|---|---|---|---|---|
| | v1 | v2 | v3 | v4 | v1 | v2 | v3 | v4 | v1 | v2 | v3 | v4 |
| Neural-LP | 74.37 | 68.93 | 46.18 | 67.13 | 52.92 | 58.94 | 52.90 | 55.88 | 40.78 | 78.73 | 82.71 | 80.58 |
| DRUM | 74.37 | 68.93 | 46.18 | 67.13 | 52.92 | 58.73 | 52.90 | 55.88 | 19.42 | 78.55 | 82.71 | 80.58 |
| RuleN | 80.85 | 78.23 | 53.39 | 71.59 | 49.76 | 77.82 | 87.69 | 85.60 | 53.50 | 81.75 | 77.26 | 61.35 |
| GraIL | 82.45 | 78.68 | 58.43 | 73.41 | 64.15 | 81.80 | 82.83 | 89.29 | 59.50 | 93.25 | 91.41 | 73.19 |
| CoMPILE | 83.60 | 79.82 | 60.69 | 75.49 | 67.64 | 82.98 | 84.67 | 87.44 | 58.38 | 93.87 | 92.77 | 75.19 |
| ConGLR | 85.64 | **92.93** | 70.74 | **92.90** | 68.29 | 85.98 | 88.61 | 89.31 | **81.07** | **94.92** | 94.36 | 81.61 |
| LogCo | 90.16 | 86.73 | 68.68 | 79.08 | 73.90 | 84.21 | 86.47 | 89.22 | 61.75 | 93.48 | 94.44 | 80.82 |
| NBFNet | **94.80** | 90.50 | **89.30** | **89.00** | **83.40** | **94.90** | **95.10** | **96.00** | – | – | – | – |
| SiaILP solo | 73.95 | 65.76 | 73.22 | 63.99 | **88.29** | **95.19** | **96.88** | **96.77** | 78.00 | 89.08 | **97.40** | **81.94** |
| SiaILP hybrid | 77.43 | 77.55 | 76.86 | 73.06 | 81.95 | 92.89 | 93.99 | 95.03 | 67.00 | 79.62 | 90.96 | 75.02 |

Table 2: The Hits@10 metric values (in %) of inductive link prediction (**entity-corrupted ranking**) on twelve dataset versions. The best score is in **bold** and the second best one is underlined.

obtained from the re-implementations by (Lin et al., 2022), whereas the relation-corrupted Hits@1 scores of TACT are still obtained from the initial paper (Chen et al., 2021). R-GCN is only used as relation-corrupted Hits@3 baselines, obtained from (Liu et al., 2021). Performances of the remaining models are obtained from their initial papers.

The AUC-PR results for the selected models are in Table 1, while their entity-corrupted Hits@10 performances are shown in Table 2. Here, "SiaILP solo" refers to using only the subgraph-based model, while "SiaILP hybrid" combines both connection-based and subgraph-based models, with output scores averaged on each triple. One can see that our models achieve several new state-of-the-art results on inductive FB15K-237 and Nell-995 datasets, but perform moderately on the inductive versions of WN18RR. This may be due to WN18RR's more sparse relation types yet denser relation connections compared to FB15K-237 and Nell-995 (Toutanova & Chen, 2015). In this case, path-based subgraphs around different entities may be the same, and hence indistinguishable to SiaLP models. This represents a trade-off, wherein prediction accuracy is sacrificed in favor of the capacity for strict inductive link prediction. Therefore, it appears acceptable for the SiaLP models to be less competitive in certain scenarios compared to entity-involved models.

| Model | WN18RR | | | | FB15K-237 | | | | Nell-995 | | | |
|---|---|---|---|---|---|---|---|---|---|---|---|---|
| | v1 | v2 | v3 | v4 | v1 | v2 | v3 | v4 | v1 | v2 | v3 | v4 |
| Neural-LP | 54.80 | 23.60 | 3.00 | 19.50 | 7.30 | 3.60 | 3.90 | 4.10 | 5.00 | 5.70 | 3.30 | 3.20 |
| DRUM | 27.70 | 3.40 | 14.10 | 26.00 | 5.40 | 3.40 | 2.70 | 2.60 | 17.00 | 5.50 | 3.80 | 1.80 |
| GraIL | 74.90 | 62.80 | 37.70 | 61.00 | 1.60 | 1.30 | 0.30 | 1.70 | 14.60 | 1.00 | 0.70 | 1.70 |
| TACT | **99.50** | **97.80** | **85.20** | **98.20** | **74.10** | 71.70 | 72.20 | 40.90 | 77.60 | 53.30 | 35.40 | 44.40 |
| SiaILP hybrid | 85.11 | 85.26 | 75.70 | 82.44 | 70.73 | **82.64** | **82.43** | **81.04** | **85.00** | **70.38** | **65.39** | **68.81** |

Table 3: The Hits@1 metric values (in %) of inductive link prediction (**relation-corrupted ranking**) on twelve dataset versions. The best score is in **bold** and the second best one is underlined.

Then, the relation-corrupted Hits@1 and Hits@3 performances of all models are shown in Table 3 and Table 4, respectively. Here, we only apply SiaILP hybrid setting, since performances by each

| Model | WN18RR | | | | FB15K-237 | | | | Nell-995 | | | |
|---|---|---|---|---|---|---|---|---|---|---|---|---|
| | v1 | v2 | v3 | v4 | v1 | v2 | v3 | v4 | v1 | v2 | v3 | v4 |
| R-GCN | 2.10 | 11.00 | 24.50 | 8.10 | 2.40 | 3.40 | 3.50 | 3.30 | 26.00 | 0.80 | 1.40 | 3.00 |
| GraIL | 0.60 | 10.70 | 17.50 | 22.60 | 1.00 | 0.40 | 6.60 | 3.00 | 0.00 | 7.40 | 2.50 | 0.50 |
| INDIGO | 98.40 | **97.30** | **91.90** | 96.10 | 53.10 | 67.60 | 66.50 | 66.30 | 80.00 | 56.90 | 64.40 | 45.70 |
| SiaILP hybrid | **99.47** | 97.05 | 91.24 | **98.46** | **81.95** | **93.51** | **93.29** | **93.68** | **100.00** | **81.30** | **83.31** | **81.12** |

Table 4: The Hits@3 metric values (in %) of inductive link prediction (**relation-corrupted ranking**) on twelve dataset versions. The best score is in **bold** and the second best one is underlined.

single model are less satisfying. We can see that our SiaILP model performs especially well on relation-corrupted rankings, which outperforms the baselines by a great margin on the inductive versions of FB15K-237 and Nell-995. Here, our models can directly use relation embeddings in relation-corrupted ranking scenarios. To our models, this is more straightforward than in entity-corrupted ranking scenarios, where entities are indirectly represented by path-based connections or subgraphs. Hence, our models become competitive when performing on the WN18RR datasets for relation-corrupted inductive link prediction.

## 4.5 ABLATION STUDIES

| Model | WN18RR | | | | FB15K-237 | | | | Nell-995 | | | |
|---|---|---|---|---|---|---|---|---|---|---|---|---|
| | v1 | v2 | v3 | v4 | v1 | v2 | v3 | v4 | v1 | v2 | v3 | v4 |
| SiaILP basic solo | 79.04 | 78.17 | 76.39 | 70.96 | 88.03 | **94.95** | 92.75 | **95.42** | 83.58 | 87.65 | **91.22** | **81.98** |
| SiaILP basic hybrid | **84.23** | **88.54** | **83.94** | 84.50 | **88.64** | 93.39 | 92.81 | 93.20 | 76.35 | **88.20** | 89.88 | 81.03 |
| SiaILP large solo | 74.48 | 80.58 | 69.11 | 73.17 | 86.63 | **94.95** | 93.62 | 94.42 | **86.96** | 88.16 | 89.83 | 81.79 |
| SiaILP large hybrid | 75.40 | 86.95 | 77.30 | **85.64** | 82.44 | 94.71 | **93.98** | 93.85 | 77.09 | 85.37 | 90.05 | 70.91 |

Table 5: Ablation study: AUC-PR performances from four different settings of SiaILP models.

| Model | WN18RR | | | | FB15K-237 | | | | Nell-995 | | | |
|---|---|---|---|---|---|---|---|---|---|---|---|---|
| | v1 | v2 | v3 | v4 | v1 | v2 | v3 | v4 | v1 | v2 | v3 | v4 |
| SiaILP basic solo | 73.95 | 65.76 | 73.22 | 63.99 | **88.29** | **95.19** | **96.88** | **96.77** | 78.00 | **89.08** | **97.40** | 81.94 |
| SiaILP basic hybrid | **77.43** | **77.55** | **76.86** | **73.06** | 81.95 | 92.89 | 93.99 | 95.03 | 67.00 | 79.62 | 90.96 | 75.02 |
| SiaILP large solo | 60.11 | 66.21 | 50.74 | 54.09 | 82.93 | 94.56 | 94.91 | 95.86 | **82.00** | 87.18 | 97.15 | **82.08** |
| SiaILP large hybrid | 56.38 | 74.83 | 59.83 | 72.78 | 81.95 | 92.05 | 94.34 | 95.96 | 76.00 | 76.52 | 91.89 | 63.56 |

Table 6: Ablation study: Hits@10 performances from four different settings of SiaILP models.

Our ablation setting is straightforward: Instead of selecting three out-reaching paths from each entity in the subgraph-based model (*'basic'*), we randomly select six out-reaching paths (*'large'*). But the number of connecting paths between two entities in the connection-based model is always three. The models are evaluated using AUC-PR and entity-corrupted Hits@10 ranking, with solo and hybrid settings to be the same. The corresponding scores are shown in Table 5 and 6, respectively.

We can see that the best performances in general come from selecting three out-reaching paths for each entity in the subgraph-based model ('basic), whereas models with solo and hybrid settings possess different advantages. This leads to the default setting in this paper.

## 5 CONCLUSION

In this paper, we proposed path-based inductive link prediction models. We only employ relation embeddings and paths embeddings to capture the topological structure of a knowledge graph, excluding entity embeddings. We apply siamese neural network architectures to further reduce the number of parameters in our models. These designs make the size of our models be negligible compared to graph convolutional network based models, when applied to large knowledge graphs. Experimental results show that our models achieve several new state-of-the-art performances on the inductive versions of the link prediction datasets WN18RR, FB15K-237 and Nell-995.

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
