# OpenReview forum: "Inductive Link Prediction in Knowledge Graphs using Path-based Neural Networks"
_ICLR.cc/2024/Conference — ICLR 2024 Conference Withdrawn Submission_

### Official Review · Reviewer_yt8Z · 2023-10-23

**Soundness:** 2 fair
**Presentation:** 2 fair
**Contribution:** 2 fair
**Rating:** 3
**Confidence:** 5

**Summary:**

The paper proposes a new path-based model called SiaILP for inductive link prediction in knowledge graphs. The model uses lightweight siamese neural networks and only depends on relation and path embeddings, which can be generalized to new entities without fine-tuning. The authors compare their model to other embedding-based and rule-based models and show that SiaILP achieves several good performances in link prediction tasks using inductive versions of WN18RR, FB15k-237, and Nell995.

**Strengths:**

1. The proposed method is easy to understand.
2. The proposed method can sometimes outperform baselines in 12 variant datasets.
3. There are few typos.

**Weaknesses:**

1. The motivation, novelty, and contribution are unclear to me when reading this paper. I can just understand what is proposed by this method, but what problems it solves, what challenges it faces, what is novel, and what the main contribution is, are not clear.

2. Lack of discussion on related works. For one thing, the subgraph-based model should be carefully compared with enclosing subgraphs used in GraIL and CoMPILE. For another, the path-based model should be carefully compared with the path-based method NBFNet. The authors also indicate that NBFNet is very similar to the proposed model, but what are the key differences?

3. The proposed method only performs the best in a few cases. The results on FB15K-237 and Nell-995 are claimed to be SOTA, but only a few numbers are the best. The relative gain is really marginal compared with the gap to the best when SiaILP is not the best.

4. The design of relation embeddings is inspired by Word2vec, but word2vec is for words/nodes. The authors should talk more about this design.

5. Lack of complexity analysis. It is well known that the number of paths can be large between two entities, so the running time comparison and complexity analysis are essential.

**Questions:**

1. What problems of existing methods does the proposed method solve?
2. What are the challenges when designing the method?
3. Which part is original or what is the main contribution?
4. What are the key differences between the path/subgraphs with the existing methods?
5. Can you compare the running time with baseline methods in this task?

---

### Official Review · Reviewer_cRRq · 2023-10-25

**Soundness:** 2 fair
**Presentation:** 1 poor
**Contribution:** 2 fair
**Rating:** 3
**Confidence:** 4

**Summary:**

This paper proposes SiaILP, a path-based model for inductive link prediction using siamese neural networks. SiaILP leverages Bi-LSTMs to encode path information and then concatenate them as the representation of the node. It achieves comparable performance in link prediction on three datasets without fine-tuning on new entities.

**Strengths:**

This paper is well-written and easy to follow.

The proposed path-based method is interesting. The idea and design of SiaILP are clearly illustrated.

**Weaknesses:**

The use of BiLSTMs may seem outdated. Why didn’t the author consider using transformers to encode the paths?

In my opinion, SiaILP requires more resources for generating paths. However, its performance is significantly worse than state-of-the-art methods on multiple datasets.

The experiments are repetitive. The tables only differ in the evaluated metrics. In other words, it seems like the authors conducted a single experiment and presented it in multiple tables. It appears lazy.

Lastly, and most importantly, based on the current experimental results, the proposed path-based method does not show significant superiority over the other methods mentioned in the abstract.

**Questions:**

Please see Weaknesses.

---

### Official Review · Reviewer_QF3p · 2023-10-28

**Soundness:** 2 fair
**Presentation:** 3 good
**Contribution:** 1 poor
**Rating:** 3
**Confidence:** 5

**Summary:**

The paper introduces SiaILP, a path-based approach for inductive relation prediction based on the siamese neural network architecture. For each entity, SiaILP first mines paths (with some hardcoded parameters) to the neighbors or within a subgraph. For each query $(h,r,t)$, the model first selects exactly 3 relational paths between $h$ and $t$, encodes each of them through the embedding layer and BiLSTM (the “siamese” term is applied to encoding several paths with the same neural net), and uses a dot-product decoder to match with the query relation $r$. The model learns input and output relation embeddings (in addition to BiLSTM weights) and is applicable for inductive relation prediction tasks where entities at inference time can be new, but the set of relations is the same.

**Strengths:**

A relatively simple approach for relation prediction (but not for entity-level knowledge graph completion).

**Weaknesses:**

**W1.** No scientific novelty - SiaILP is a rather brute-force approach that directly mines relational paths for each triple in each inference graph and applies standard BiLSTM for encoding the paths. When there are not enough paths (less than 3), the approach resorts to mining paths within the entity’s subgraph. The authors do not provide any detail on the preprocessing time or memory requirements necessary for path mining for the benchmarked datasets. Generally, such brute-force mining of a hardcoded number of paths is always inferior to GNNs like NBFNet that capture _all_ possible paths thanks to the algorithmic alignment between GNNs and dynamic programming (such as the Bellman-Ford algorithm in NBFNet).

**W2.** There is no motivation behind the design choices in SiaILP, e.g., why the model takes exactly 3 paths, why the decoder is a concatenation-MLP which is permutation-variant (i.e., a different order of the same aggregated 3 paths would return different results), why is there a need for an RNN. All the components seem to be chosen ad-hoc and leave an impression of a technical report of a quickly assembled ML course project.

**W3.** The state-of-the-art claims are rather exaggerated - SiaILP only shows competitive (to the 2021 baselines) performance in the easiest relation prediction task $(h, ?, t)$ when there are at most 200 relations to rank in the benchmarked splits. Relation prediction is a rather trivial task that usually supports the main entity prediction $(h, r, ?)$ task, the actual KG completion. When it comes to the KG completion, SiaILP is evaluated on the toy setup of 50 random negatives (originally chosen because of the excessive computational complexity of GraIL) whereas modern approaches use evaluation on the full entity set (including NBFNet, RED-GNN [1], ReFactorGNNs [2]). Even in that toy entity ranking setup, SiaILP is mostly behind the old baselines (sometimes by a large margin).

Following that, I would recommend reporting evaluation results on the full entity ranking (for both head and tail prediction) task. I suspect that due to the path mining and aggregation mechanism, all-entity ranking for SiaILP would take an enormous time that would render the approach inapplicable for any graph larger than those small GraIL splits.

Overall, I do not see enough theoretical or experimental merits that would support the publication at ICLR.


**Writing comments:**
The paper contains rather strange wording in some parts:
* Abstract: “typologies” -> topologies.
* Abstract: “rules … lack generosity” -> lack generalization?
* Experimental results: “... if the existing structure reaches an optimism” -> optimum? Optimality?


**References:**
[1] Zhang et al. Knowledge graph reasoning with relational digraph. WebConf 2022.
[2] Chen et al. ReFactor GNNs: Revisiting Factorisation-based Models from a Message-Passing Perspective. NeurIPS 2022.

**Questions:**

N/A

---

### Official Review · Reviewer_GEeR · 2023-10-31

**Soundness:** 2 fair
**Presentation:** 1 poor
**Contribution:** 1 poor
**Rating:** 3
**Confidence:** 3

**Summary:**

This paper focuses on inductive link prediction task, which is a crucial research area in knowledge graph. The authors design a path-based model using light-weight siamese neural networks to generalize to new entities, which only depends on relation and path embeddings. Experimental results demonstrate this model can achieve good results on some inductive version knowledge graph datasets.

**Strengths:**

S1. The proposed model is strictly inductive, because it does not rely on entity embeddings and can be directly applied to new entities without fine-tuning.

S2. The related works are presented comprehensively, covering transductive link prediction and inductive link prediction.

**Weaknesses:**

W1. The novelty of this work is limited. NBFNet[1] also takes advantages of path-based model for inductive link prediction. The proposed model of this paper just designs a subgraph-based module to enhance connection-based module.

[1] Zhu, Zhaocheng, et al. "Neural bellman-ford networks: A general graph neural network framework for link prediction." Advances in Neural Information Processing Systems 34 (2021): 29476-29490.

W2. The biggest concern of the reviewer is the experimental results. In table1 and 2, the proposed model SiaILP is hard to achieve state-of-the-art performance compared to existing baselines. This is not consistent with the authors' claims in the previous part. And in table 5 and 6, the  hybrid version is also hard to outperform solo version, so it can not prove the effectiveness of the proposed siamese network.

W3. The presentation of the method is not very clear. In section3.3, they said "The source and target paths share the same input embedding matrix..." and "Note that there is no topological difference between source entity subgraph and target entity subgraph". But in fact, the input embedding matrix and subgraph topology of source and target entities are different.

W4. Some claims are inaccurate. In Section4.2, "Afterwards, we train the connection-based model using contrastive learning...", however contrastive learning refers to pulling positive samples and pushing negative samples, which is not suitable to describe negative sampling.

W5. The paper mentions that the proposed model is light-weight, but there is no study about efficiency and complexity analysis.

**Questions:**

Q1. It is better to report the statistics of the datasets for self-contained presentation, although they are summarized in other papers.

Q2. Please explain W3 above.

Q3. Please explain why only three paths are considered in the proposed method.

---

### Author Response · Authors · 2023-11-11

Thank you for the suggestions and feedback!